# The Role of Partial Enteral Nutrition for Induction of Remission in Crohn’s Disease: A Systematic Review of Controlled Trials

**DOI:** 10.3390/nu14245263

**Published:** 2022-12-09

**Authors:** Lucía González-Torres, Ana Moreno-Álvarez, Ana Estefanía Fernández-Lorenzo, Rosaura Leis, Alfonso Solar-Boga

**Affiliations:** 1Department of Pediatrics, Hospital Materno Infantil Teresa Herrera, Área Sanitaria A Coruña-Cee, 15009 A Coruña, Spain; 2Pediatric Gastroenterology, Hepatology and Nutrition Unit, Hospital Materno Infantil Teresa Herrera, Área Sanitaria A Coruña-Cee, 15009 A Coruña, Spain; 3Unit of Pediatrics Gastroenterology, Hepatology and Nutrition, Pediatrics Department, Hospital Clínico Universitario de Santiago, Pediatrics Nutrition Group-IDIS, CiberOb, 15706 Santiago de Compostela, Spain

**Keywords:** inflammatory bowel disease, Crohn’s disease, nutritional therapy, enteral nutrition, diet

## Abstract

Exclusive enteral nutrition (EEN) is recommended as a first-line therapy to induce remission of Crohn’s disease (CD) and is considered as effective as corticosteroid treatment. However, the dietary restriction causes lack of adherence and poor tolerance to the therapy. Partial enteral nutrition (PEN), which allows for the ingestion of some food, could be a better tolerated alternative, but it is unknown whether it is as effective at inducing CD remission as EEN. The aim of this systematic review is to analyze the available evidence on PEN as a remission induction therapy in CD. A literature search was conducted using the MEDLINE (via PUBMED) and Cochrane Library databases following the preferred reporting items for systematic reviews and meta-analyses (PRISMA) guidelines. Clinical trials in pediatric and adult patients were included. The risk of bias was assessed following the Cochrane Collaboration methodology. The selected studies showed variable but high response rates to PEN and EEN. Limitations regarding the wide heterogeneity between the studies included in this review should be considered. Although more studies are needed, according to our results, PEN combined with a highly restrictive diet seems to be as effective as EEN in inducing remission of CD.

## 1. Introduction

Crohn’s disease (CD) is a chronic inflammatory disease of unknown etiology that affects the gastrointestinal tract and typically presents in flares, with periods of activity and remission [1]. The goal of disease treatment is to induce and maintain clinical remission and achieve mucosal healing to prevent complications and modify, if possible, the progressive course of the disease [2,3]. Several therapeutic options exist for both the induction and maintenance of remission. The latest recommendations of the ECCO-ESPGHAN [4] include pharmacological treatments such as mesalazine, local and systemic steroids, thiopurines, methotrexate and biological therapies. Despite advances in pharmacological treatment and high response rates, medical therapies are associated with high morbidity due to side effects resulting from immunosuppression [5]. Exclusive enteral nutrition (EEN) therapy consists of the administration of a liquid formula as the only food source for six to eight weeks with the exclusion of any other oral intake except water. Depending on their composition, the formulas can be polymeric (45–60% carbohydrates, 15–20% whole proteins and 30–40% fat), semi-elemental (oligopeptides, dipeptides or tripeptides and medium-chain triglycerides) and elemental formulas (fully hydrolyzed macronutrients such as amino acids and simple sugars with low fat content) [6]. EN is recognized as a therapeutic option that avoids the side effects of pharmacological treatment and leads to mucosal healing. Its benefits have been known since the 1980s [7], and from that time, multiple studies and meta-analyses have shown EEN to be as effective [8,9,10], or even better [11,12], than corticosteroid therapy. EEN is considered to induce clinical remission, and mucosal and transmural healing, and has a positive effect on growth, bone health [13], nutritional status [14] and quality of life [15]. The current position of the ESPGHAN [16] and the Porto Inflammatory Bowel Disease group [17] regarding nutritional therapy of CD is in favor of EEN, which is indicated as a first-line therapy for the induction of remission in children with active luminal CD. Regarding the type of formula used, the evidence does not show significant differences [6] but, considering the reduced palatability, the risk of early withdrawal and high cost associated with elemental diets, first-line therapy with polymeric formula seems justifiable [16]. However, in adults, the latest therapeutic management guidelines still do not recommend EEN as a first-line therapy in the treatment of active CD [5]. The main barriers to the use of EEN as a therapy are tolerance and adherence to treatment [9,18,19], which sometimes lead to administration of a nasogastric tube or abandonment of therapy [14,17]. In addition, the important dietary restriction implies a great effort and sacrifice for both the patient and his family, which also affects the psychosocial sphere [20]. One way to improve adherence to this therapy would be to allow for the introduction of a small amount of food. Partial enteral nutrition (PEN) provides a percentage of the daily caloric requirements using the enteral formula combined with other food intake [21]. Early studies into this dietary option showed poor results for the induction of remission [22]. However, in recent years, new research on the role of diet in the pathogenesis of inflammatory bowel disease [7,23] has meant that partial enteral therapy combined with an anti-inflammatory diet such as the Crohn’s disease exclusion diet (CDED) [24] or Crohn disease treatment-with-eating diet (CD-TREAT) anti-inflammatory diets [25] are back on the table. The CDED diet is designed to exclude dietary components that adversely affect intestinal permeability, the microbiome or the immune system [24]. It avoids foods such as gluten, dairy products, baked goods and breads, animal fat, processed meats, products containing emulsifiers, canned goods and all packaged products. The CD-TREAT diet [25] takes a different approach by attempting to recreate the composition and effects of an enteral formula using solid foods and excluding certain dietary components, such as gluten, lactose or alcohol. Despite these innovative options, the evidence regarding the use of PEN is scarce and, at present, the ESPGHAN Inflammatory Bowel Disease Porto Group does not recommend it as a stand-alone therapy to induce remission of CD [17]. The aim of this systematic review is to analyze the available evidence on the efficacy of PEN as remission induction therapy in CD.

## 2. Materials and Methods

This review was conducted following the preferred reporting items for a systematic review and meta-analysis (PRISMA) checklist [26]. To structure the review process, the population, intervention, comparison, and outcome (PICO) model was followed [27] (Table 1).

### 2.1. Search Strategy

A systematic literature review was conducted to identify relevant publications on the efficacy of PEN as a primary therapy for the induction of remission in CD. For this purpose, the main databases MEDLINE (Pubmed) and Cochrane Library were searched.

The search formula used in the MEDLINE database was: ((crohndisease [MeSHTerms]) OR (crohn s disease [MeSHTerms]) OR (crohn sdisease [MeSHTerms]) OR (crohnsdisease [MeSHTerms]) OR (inflammatoryboweldisease [MeSHTerms]) OR (inflammatoryboweldiseases [MeSHTerms])) AND ((enteral nutrition [MeSHTerms]) OR (enteral feeding [MeSHTerms]) OR (feeding, enteral [MeSHTerms]) OR (partial enteral nutrition [Title/Abstract]) OR (exclusive enteral nutrition [Title/Abstract])). The search strategy in the Cochrane Library database was #1 EXCLUSIVE ENTERAL NUTRITION, #2 partial enteral nutrition, #3 MeSH descriptor: [Crohn Disease] explodeall three, #4 MeSH descriptor: [Enteral Nutrition] explodeall three, #5 MeSH descriptor: [InflammatoryBowelDiseases] explodeall three, #6 (#3 OR #5) AND (#4 OR #2 OR #1).

### 2.2. Inclusion and Exclusion Criteria

Interventional analytical studies (randomized or non-randomized clinical trials) published in Spanish or English between January 2011 and December 2021 were considered. Studies with patients of any age with a diagnosis of active CD, as defined by an index of clinical disease activity, were included. Observational and descriptive analytical studies, preclinical studies, preprint studies, clinical guidelines, reviews and protocols were excluded. Gray literature was not included.

### 2.3. Type of Intervention

We considered studies that analyzed the administration of PEN (administration of enteral formula providing less than 100% of daily calories combined with food in the form of a free diet or specific diets) as the main therapy for the induction of remission of active CD and compared this with EEN.

### 2.4. Types of Outcome Measures

We included studies whose primary or secondary study outcomes were:-Clinical response or remission: assessed with some index of disease activity such as the Crohn’s Disease Activity Index (CDAI), the Harvey–Bradshaw Index (HBI), the Short Inflammatory Bowel Disease Questionnaire (SIBDQ), and the Lehmann score [5] for adults or the Pediatric Crohn’s Disease Activity Index (PCDAI) or Weighted Pediatric Crohn’s Disease Activity Index (wPCDAI) [28] in pediatric patients.-Endoscopic remission or mucosal healing: assessed by endoscopic study and using endoscopic scoring indices such as the Crohn’s Disease Endoscopic Index of Severity (CDEIS) or the Simple Endoscopic Score for Crohn’s Disease (SES-CD) [29].-Mucosal healing: visual or according to the SES-CD index (SES-CD = 0).-Analytical response of remission: with inflammatory markers such as C-reactive protein (CRP), erythrocyte sedimentation rate (ESR) or fecal calprotectin [16].-Anthropometric assessment: weight, height, body mass index (BMI).-Nutritional status and analytical parameters: insulin-like growth factor type 1 (IGF-1), albumin, and prealbumin.-Adherence and tolerance to treatment.

### 2.5. Selection of Studies

Two authors (L.G.T. and A.F.L.) independently selected the articles included in the review by searching databases. In cases where consensus was not reached, A.M.A. and A.S.B. acted as referees. The articles identified were assessed by title and abstract, and those selected for inclusion were read in their entirety.

### 2.6. Data Extraction

The following data were extracted from the selected articles: study details (author, year of study, study design, sample size, duration of follow-up); participant details (age, inclusion/exclusion criteria, clinical or endoscopic disease activity indices, type of intervention (dietary intervention, mode, duration, type of formula), primary and secondary objectives, and outcomes. Any discrepancies were refereed by the remaining authors.

### 2.7. Quality Assessment and Bias Analysis

A bias assessment scale was used according to each type of study. “Rob-2: Version 2 of the Cochrane risk-of-bias-assessment tool for randomised trials The Cochrane Risk of Bias-assessment tool for randomised trials” [30] for randomized clinical trials. Inherent limitations and shortcomings of the studies were also considered to complement these tools. Two assessors independently assessed the risk of bias in each study. We assessed selection bias (random sequence generation, allocation concealment), performance bias (blinding of participants and personnel), detection bias (blinding of outcome assessment), attrition bias (incomplete outcome data), reporting bias (selective reporting) and any other form of bias. The risk of each type of bias was classified as low, high or unclear in those cases in which insufficient data were reported. In cases of lack of consensus, A.M.A. and A.S.B. acted as referees.

## 3. Results

The MEDLINE database search yielded 397 records, and the Cochrane Library yielded 47 records. In total, 444 articles were included in the bibliographic search. They were selected by title and reviewed by abstract, and 12 articles were selected for full-text review. Finally, five articles [31,32,33,34,35] that matched the selection criteria were selected. The results of the search are summarized in the flow chart in Figure 1 (Figure 1) and the reasons for the exclusion of articles that were read in full but not included in the review [18,36,37,38,39,40,41] are summarized in Appendix A.

### 3.1. Risk of Bias

The quality and risk of bias were determined using the Cochrane “Rob-2: Version 2 of the Cochrane risk-of-bias-assessment tool for randomised trials” [30] and are summarized in Table 2 of the Appendix A (Appendix A). All the analyzed studies had an inherent risk of bias due to the absence of blinding, which is due to the nature of the intervention itself (exclusive or partial administration of enteral nutrition). The studies published by A. Levine et al. [31], R. Sigall-Boneh et al. [32] and Yanai et al. [33] show a low risk of bias and the studies published by D. Urlep et al. [34] and C. Wall et al. [35] present a high risk of bias, mainly due to the absence of randomization in both cases.

### 3.2. Study Characteristics and Type of Intervention

Table 2 summarizes the inclusion and exclusion criteria of the selected studies, which were not homogeneous among the studies. All of them required the diagnosis of active CD but the level of activity was assessed using different scales. The studies by A. Levine et al. [31] and Sigall Boneth et al. [32] included children with mild or moderate active CD (defined by PCDAI > 10 points and ≤40 points with analytical evidence of inflammation -CRP > 5–6 g/dL, fecal calprotectin > 200 mcg/g ESR > 20 mm/h–). The D. Urlep et al. study [34] included children with a diagnosis of active disease by PCDAI > 10 or endoscopic SES-CD index > 3. Young adult patients were included in the study by C. Wall et al. [35] and H. Yanai et al. [33], with a diagnosis of CD with at least ileal involvement defined by elevated inflammatory markers [31], endoscopy or radiology [33]. All the studies excluded patients on concomitant systemic corticosteroid therapy [31,32,33,34,35] or budesonide greater than 3 mg [33], and previous or current use of biologics was not allowed in some of them [31,32,33]. The perianal involvement or penetrating disease was an exclusion criterion in all studies except that of C. Wall et al. [35]

The main characteristics of the five selected trials are summarized in Table 3. Regarding nutritional intervention, all the clinical trials compared PEN, as the induction therapy in CD, to another dietary treatment: EEN [31,32,34,35] or CDED [33]. Non-exclusive enteral nutrition was used, with different schedules: the formula provided a different number of daily calories—from 25–50% [31,32,33] to 75% [34]—and was combined with different diets: CDED [31,32,33], an anti-inflammatory diet for CD (AID-CD), designed based on the former [34], with one free meal each day [35]. Enteral nutrition was provided by a polymeric normocaloric normoproteic formula. Time taken to assess results ranged from 6 [32,34] to 24 weeks [33].

### 3.3. Clinical and Analytical Response

Outcome measures were evaluated differently in each article: a clinical response as a PCDAI decrease > 12.5 points [31] or >15 points [34] in studies with children, or HBI decrease if the cohort was adults [33,35]. In the five selected studies, high rates of clinical improvement in both EEN and PEN were observed, with no statistically significant superiority of one therapy over the other. The results regarding PEN’s efficacy are shown in Table 4.

The study by A. Levine et al. [31], observed a significant decrease in median PCDAI with treatment in both groups (PEN from 25 points to 2.5 and EEN from 27.5 to 5 points (0–10), both *p* < 0.001). However, no statistical differences were found in clinical response (PEN 85% vs. EEN 85.3%, *p* = 0.36) or clinical remission rates with different nutritional therapies (PEN 80% vs. EEN 73.5%, *p* = 0.51). Similar results were observed in the study by D. Urlep et al. [34], and in pediatric patients, where high clinical response rates were observed, with a PCDAI ≥15 points decrease in PEN for 100% of patients (11/11) vs. 90.9% (10/11) in EEN (10/11), although without significant differences between the two (*p* = 0.999). There were also no differences in remission rates with exclusively enteral nutrition or combined (PCDAI < 10 points: EEN 9/13 (69.2%) vs. PEN 9/12 (75%%), *p* = 0.999).

Improvements in the activity score HBI were evaluated by C. Wall et al. [35]. At the end of the two weeks of EEN induction, high clinical response rates were observed, with the HBI decreasing from 5 to 3 points (*p* = 0.003). During the following 6 weeks of therapy, both groups decreased their HBI, although this was only statistically significant in the EEN group and there were no significant differences between the two groups. In the study by Yanai et al. [33], the number of patients who achieved clinical remission at week 6 (HBI score < 5) and sustained remission at week 24 was greater in the PEN + CDED group than in the exclusive CDED group, although this difference was not statistically significant.

All studies showed significant improvements in analytical parameters in both intervention groups, with no significant differences between the PEN group and the other group. In the study by A. Levine et al. [31] a significant improvement in median CRP was observed in the first six weeks of treatment, with a decrease from 23.6 mg/L to 5.0 mg/L in the PEN group and from 24.0 mg/L to 4.1 mg/L in the EEN group (both *p* < 0.001). No significant differences in CRP normalization were observed between the two groups (*p* = 0.69). In the study by C. Wall et al. [35], a significant decrease in analytical parameters was observed in the first two weeks of EEN treatment in both groups, with CRP decreasing from a median of 10.0 to 5.0 mg/L (*p* = 0.005). In the following six weeks of treatment, both groups maintained the improvement in CRP, although there were no differences between the two groups. A decrease in CRP with treatment was also observed by Yanai et al. [35], but without differences regarding the use of PEN with CDED.

Finally, R. Sigall-Boneh et al. [32] analyzed disease remission, clinical response and analytical parameters at week three, the only study included in this review that evaluates such an early response. They did not observe significant differences in clinical response rates (EEN 85% vs. PEN 82%, *p* = 0.71) or in clinical remission (EEN 64.7% vs. PEN 61.5%, *p* = 0.78). They did, however, observe an improvement in CRP in both groups, with almost 50% of patients reaching normalization of CRP at week three.

### 3.4. Fecal Calprotectin and Mucosal Healing

Regarding fecal calprotectin, studies showed mixed results. In the study by A. Levine et al. [31], a significant drop in fecal calprotectin was observed in both treatment groups at week six, with no statistically significant differences (*p* = 0.43) between the two groups (PEN calprotectin drop from 3126 mcg/g to 1744 mcg/g (*p* = 0.002) and EEN group from 2647 mcg/g to 1021 mcg/g (*p* = 0.011)). However, it is noteworthy that, in the following six weeks of treatment, calprotectin continued to decrease in the PEN + CDED group (1744 to 732, *p* = 0.22), while in the group that had received EEN and introduced a free diet, a rebound of median calprotectin from 1021 mcg/g to 1589 mcg/g was observed, although this was not statistically significant (*p* = 0.36).

In the study by C. Wall et al. [35], a decrease in median fecal calprotectin from 927 mcg/g to 674 mcg/g (*p* = 0.028) was observed in both groups in the first two weeks of treatment. However, despite the initial improvement, in the following six weeks, both EEN and PEN showed no further improvement in fecal calprotectin concentration. In the PEN group, an increase in calprotectin was observed in 55% of patients (5/9) (*p* = 0.91).

A significant decrease in CRP, ESR and calprotectin was also observed in the study by D. Urlep et al. [34] after six weeks of treatment, although without significant differences between the two groups. Of note, there was a greater decrease in calprotectin in the PEN group (67%, from 426.5 mcg/g to 138.2 mcg/g, *p* < 0.001) compared to the EEN group (45.7%, from 381.1 mcg/g to 206.9 mcg/g, *p* = 0.009), although no statistically significant differences (*p* = 0.064) were observed between them. In addition, this study evaluated endoscopic response to nutritional therapies in children, without observing statistically significant differences between the two in endoscopic response: EEN 7/11 (63.6%) vs. PEN 10/11 (90.9%), *p* = 0.311, endoscopic remission (SES-CD ≤ 2 points) EEN and PEN 5/11 (45.5%), *p* > 0.99 and mucosal cure (SES-CD = 0) EEN 5/11 45.5% vs. PEN 3/11 27.3%, (*p* = 0.659).

Calprotectin and mucosal healing were also evaluated by Yanai et al. [33], with a decrease in levels of fecal marker at week 12: from 229.0 mgc/g to 104.1 mcg/g in the PEN + CDEC group and from 294 mcg/g to 97.3 mcg/g in the exclusive CDED group. This study evaluated endoscopic response for a longer time period, performing control colonoscopies at six months. A decrease in the SES-CD score from baseline to week 24 was observed, with the number of patients who achieved endoscopic remission being higher with combined PEN + CDED (8/15, 53%) than with CDED alone (6/13, 46%), although this was not statistically significant.

### 3.5. Tolerance, Adherence and Nutritional Status

Assessing tolerance and adherence to nutritional therapy was the main objective of the study by A. Levine et al. [31], which observed a higher tolerance of PEN 39/40 (97.5%) vs. EEN 28/38 (73.7%), *p* = 0.002 (95% CI: 9–38.6%). Less therapeutic adherence to PEN with CDED was observed in studies with adult patients. Yanai et al. [33] found lower compliance with the dietary measures at week six, with 63% in the PEN + CDED group and 86% in the CDED alone group.

Adherence to treatment was also assessed in the other studies, although not as a primary endpoint, and no significant differences were found between the two groups. The main reasons for loss of adherence were intolerance to formula, non-response or disease exacerbation, surgery and non-compliance with the protocol.

Nutritional status showed heterogeneous results. C. Wall et al.’s study [35] was the only one where non-significant weight loss was observed with both therapies: in the EEN group, the mean BMI decreased from 23.7 to 23.3 kg/m^2^ (*p* = 0.01), and in the PEN group, a modest but not significant decrease in BMI was also observed. Nutritional parameters such as IGF-1 and albumin improved during treatment in the EEN group (*p* = 0.043 and *p* = 0.047, respectively), and in the PEN group, (IGF-1—*p* = 0.17 and albumin *p* = 0.5). In the study by A. Levine et al. [31], a significant improvement in weight z-score was observed at six weeks in both groups (the PEN group weight z-score increased from 0.91 ± 1.2 to 0.64 ± 1.05 (*p* < 0.001), and in the EEN group, the z-score increased from 0.92 ± 1.17 to 0.63 ± 1.1 (*p* < 0.001), without a significant difference between groups (*p* = 0.476). In the study of D. Urlep et al. [34], weight and BMI remained stable in both groups after six weeks of treatment and a significant improvement was observed in albumin levels in both groups: EEN 40.0 g/dL to 43.2 g/dL (*p* = 0.049) and PEN 39.8 g/dL to 43.3 g/dL (*p* = 0.012), with no statistically significant difference between the two groups (*p* = 0.88). Yanai et al. [33] showed that the use of enteral nutrition implies a higher caloric intake. Thus, the PEN + CDED group presented greater weight gain at weeks 6 and 24 compared to CDED alone, although there were no differences between the groups and, in the latter group, two patients lost weight intentionally.

## 4. Discussion

Exclusive enteral nutrition is currently the first-line therapy recommended by the ESPGHAN to induce remission of active Crohn’s disease [16]. It has shown superiority over corticosteroid therapy in numerous studies and has high clinical, analytical and endoscopic remission rates, as well as showing benefits in mucosal healing and improved nutritional status [9]. One of its main limitations is the lack of adherence to treatment and the patients’ difficulties maintaining an exclusively liquid diet for 6–8 weeks [9,18,19]. PEN could make up for these limitations by allowing for patients to ingest a certain percentage of selected foods per day that could improve both their tolerance to the therapy itself and their quality of life. This type of nutritional therapy has not been sufficiently studied and there is scarce evidence on the efficacy of PEN for remission induction at present, as it is only recommended as a maintenance therapy [17]. The studies selected in this systematic review compare PEN with EEN and have similar results, with high response rates and clinical remission with both therapies, showing that PEN is at least as effective as EEN for inducing remission of active Crohn’s disease. These findings differ with the only randomized clinical trial comparing PEN and EEN for CD treatment, published prior to 2011, by Jonhson et al. [22]. This study compared a group of patients receiving PEN with a free diet and EEN, showing worse remission rates in the PEN group (PEN 15% vs. EEN 42%; *p* = 0.035). Several factors may have influenced these results, such as the concomitant use of corticosteroids or their combination with a free diet. In contrast to this study, and in line with the results of this review, the study published by Gupta et al. in 2013 [42] is one of the first studies showing PEN to be an effective therapy in remission induction, and with better tolerance than EEN. It is a retrospective study in children with Crohn’s disease flare-ups with poor response to treatment with corticosteroids or biologics, for whom 80–90% PEN was started with a 10–20% free diet for 6–12 weeks. It was observed that PEN was effective for the induction of remission and improvements in analytical and nutritional parameters.

Similarly, a descriptive study without a control group, published by Sigall-Boneh et al. in 2014 [24], analyzed the combination of PEN with a Crohn’s disease exclusion diet (CDED) as remission induction therapy, observing high clinical and analytical response rates in both adult and pediatric patients. Recent observational studies postulate the exclusive CDED diet as a therapeutic option to induce remission, without supplementation with an enteral formula. With promising initial results in adult patients [43], clinical trials such as the one recently published by H. Yanai et al. [33], included in this systematic review, showed that CDED has a high clinical response rate for the induction of remission.

The most probable key factor in the response to partial nutritional therapy is the type of diet that accompanies this. To date, the mechanism by which EEN is effective is uncertain. It is not clear whether it is through the provision certain nutrients or the exclusion of others. There are several hypotheses that attempt to explain the efficacy of EEN in the induction of remission; these consider that, by eliminating dietary components, the proinflammatory response is modified, the restoration of the epithelial barrier is promoted, and the intestinal microbiome is restored [44]. A recently published randomized clinical trial [45] evaluated the changes in fecal metabolites induced by CDED + PEN and EEN and their relationship with remission. Both therapies showed changes in metabolites such as kynurenine, ceramides or amino acids, with a different metabolomic profile between both groups. The CDED + PEN group that achieved sustained remission showed persistent metabolome changes. However, in the EEN group, when free diet was allowed to be introduced, the changes returned to baseline values. Non-responder patients with EEN, did not show significant changes in fecal metabolites. This is consistent with the theory that the type of diet plays a role in the response to nutritional therapy. In this scenario, PEN could be contemplated and combined with an exclusion diet that eliminates these “noxious foods” that promote proinflammatory issues.

A matter of debate regarding the nutritional treatment of Crohn’s disease is whether the association of enteral nutrition with a specific diet is necessary to achieve remission. The study by Yanai et al. [33] compares the efficacy of CDED depending on whether it is associated with PEN. In their work, they found no significant differences in the two groups in measures of effect. However, the group associated with PEN presented more favorable results in terms of response at week six (73.7% vs. 66.7%), sustained remission at week 24 (63.2% vs. 38.2%) and endoscopic response (42.1% vs. 28.6%). A limited number of patients in the study (fewer than 20 in each group that completed 24 weeks of treatment) may determine statistical significance. Recently, a retrospective real-world study [46] observed similar efficacy in the induction of remission in children with CD treated with CDED + PEN in comparison with EEN, along with greater weight gain. Nevertheless, the 80% of CDED + PEN-treated patients had previously received 1–2 weeks of EEN, so larger studies should be carried out to assess nutritional therapy, which may be promising in certain groups of patients.

In this review, all the studies show high response rates and clinical and analytical remission, with both EEN and PEN with no significant differences between both therapies. However, regarding changes in fecal calprotectin, the results are mixed. Studies by A. Levine et al. [31], Sigall-Boneh et al. [32] and D. Urlep el al. [34] show a greater decrease in calprotectin in the PEN groups, but not the study by C. Wall et al. [35], which observed a rise in calprotectin in the group receiving PEN. However, if the dietary intervention in each study is analyzed in depth, in the study by A. Levine et al. [31] fecal calprotectin drops equally in both groups in the first six weeks of treatment; however, in the following six weeks, calprotectin continued to drop (1744 to 732 mcg/g, *p* = 0.22) in the group that maintains PEN with CDED, although a rebound is observed in the EEN group with the reintroduction of a free diet (combined with PEN 25%). This is the same as the study by C. Wall et al. [35], who observed a greater decrease in calprotectin in the group maintaining EEN and an increase in calprotectin in 55% (5/9) of patients in the PEN group (*p* = 0.91) introducing a free diet. In both cases, the key in the increase in calprotectin seems to lie in the reintroduction of a free diet, since calprotectin values remain low not only in the groups maintaining EEN but also in patients with PEN with a specific exclusion diet. These data support the hypothesis that the efficacy of nutritional therapy is associated with the exclusion of certain components of the usual diet. In this line, Logan M et al. [47], prospectively observed that after the decrease in calprotectin induced by EEN therapy, levels increased again at the time of reintroduction of normal solid food. A nonspecific diet accompanying PEN was also associated with less favorable results in the study by D. Lee et al. [48]. This prospective study in children with active CD with three treatment groups (anti-TNF, EEN and PEN), showed higher clinical response rates in the EEN (88%) and anti-TNF (84%) vs. PEN (64%) group, (*p* = 0.08), as well as worse results in terms of normalization of fecal calprotectin (calprotectin < 250 mcg/g) in the PEN group (14%) vs. EEN (45%) and anti-TNF (62%) (*p* = 0.001). It should be noted that the PEN group received a free diet and not an exclusion diet, again showing that the free diet is related to worse clinical and analytical outcomes.

The findings observed in fecal calprotectin are also observed in the intestinal microbiome. In the study by A. Levine et al. [31], changes in the microbiome were evidenced after the first six weeks in both groups (EEN and PEN with exclusion diet), showing a decrease in specific species of proteobacteria, such as *Haemophilus* spp, *Veillonella* spp, *Anaerostipes* spp. or *Prevotella* spp. and an increase in *Firmucutes* spp. However, the EEN group showed a rebound effect in the composition of the microbiome (particularly Proteobacteria) after the introduction of free diet, which returned to the initial state. These changes occurred in parallel with the rebound of calprotectin and CRP in the EEN group between week 6 and week 12.

Observing that the clinical and analytical response rates of PEN are similar to those of EEN, in this systematic review, one of the advantages that the former appears to have over the latter is the degree of tolerance and adherence to therapy. This was the main objective of the study by A. Levine et al. [31], included in our review, which observed significantly greater tolerance in the PEN group (97.5% vs. 73.7%, *p* = 0.002). The fact of being able to share one meal a day with their families or friends, even if it is a restrictive diet, allows for them to maintain a certain degree of social life and to favor quality of life standards. In addition to the abandonment of EEN due to reasons such as intolerance to taste, nausea, vomiting or diarrhea, the age of the patient is an important factor in the maintenance of therapy. Only one of the studies of this referral assesses age as an element that may influence success with therapy [32], observing a greater number of losses in adolescent patients, regardless of whether they received EEN or PEN. Adolescents are likely to be a particularly vulnerable group, with higher rates of treatment failure for different reasons. In addition to age, another important factor in adherence to treatment could be the time of disease progression; however, this was not analyzed in the studies included in this review, nor were socioeconomic or cultural factors considered. A 2020 study by Wall CL et al. [18] analyzed how patient personality can influence adherence to partial and EEN therapy using a questionnaire, observing that more rigorous patients better adhere to strict therapies such as EEN, and patients classified by the questionnaire as less rigorous complied better with a partial nutritional therapy that allowed for one meal a day. For all these reasons, to predict good adherence to nutritional treatment, it seems important to evaluate and individualize certain elements such as the patients own personality, age, family support network or economic level. In fact, the establishment of prognostic factors of response to treatment, such as age, disease location, disease behavior, presence of growth retardation, possible side effects [14] or even genetic factors [49], will be key to the success of the CD treatment, allowing for nutritional treatment and specific medical therapies to be individualized and targeted [50].

A notable limitation of this review is that the studies are not very homogeneous in terms of the type of diet that was combined with PEN, since some used CDED [31,32,33], one used an adapted exclusion diet [34] and another a free diet [35]. This reflects the multiple dietary options being tested in inflammatory bowel disease, such as CDED, CD-TREAT, food influence on the intestinal microbiota diet (FIT), specific carbohydrate diet (SCD), the anti-inflammatory diet (BD-AID), or the low-fermentable oligosaccharide, disaccharide, monosaccharide and polyols diet (low FODMAP) [51]. It is important to note that the most promising results in favor of partial nutritional therapy are observed in studies that develop very demanding partial nutritional interventions, performing CDED exclusion diets or allowing for only one meal per day. However, all the studies included in this review have an inherent risk of bias due to the absence of blinding due to the nature of the intervention itself, with two studies being classified as having a high risk of bias according to the “RoB-2” risk of bias analysis tool [30]. Another important limitation is that the study with the highest weight in terms of number of patients [31] only included patients with mild-moderate disease (PCDAI < 40 points), which may influence the better results obtained with PEN. It should also be mentioned that only two studies [3,33] in this systematic review evaluated endoscopic response, and more studies are needed in this regard.

## 5. Conclusions

The studies analyzed in this systematic review show high clinical and analytical response rates in CD treatment using both EEN and PEN, with no statistically significant differences between the two. Nutritional therapies that allow for not only enteral nutrition alone but also some kind of food appear to be as effective as EEN for inducing CD remission. However, better clinical and analytical responses are observed with scheduled treatments that combine an enteral formula with specific diets or with a normal free diet but at a low percentage. Based on our findings, although PEN is shown to be a promising therapy for the induction of remission in CD, more clinical trials studying this approach focusing on different specific diets combined with PEN are needed.

## Figures and Tables

**Figure 1 nutrients-14-05263-f001:**
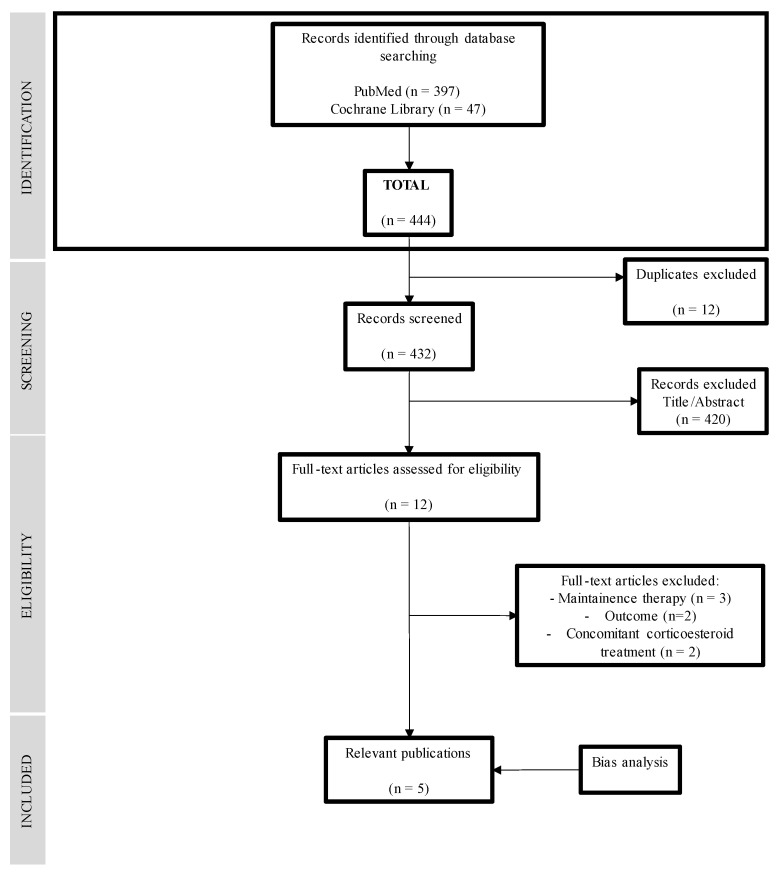
Flowchart of the search strategy and results.

**Table 1 nutrients-14-05263-t001:** Population, Intervention, Comparison, and Outcome (PICO) criteria [27] for the inclusion of studies on the efficacy of partial enteral nutrition for induction of Crohn’s disease remission.

	Inclusion Criteria
Population	Patients of any age with a diagnosis of active CD, as defined by an index of clinical disease activity.
Intervention	The administration of PEN as the main therapy for induction of remission of active CD.
Comparison	EEN or any other control group.
Outcome	- Clinical response or remission: assessed with some index of disease activity such as the CDAI, HBI (Harvey–Bradshaw index), the Lehmann score for adults; or the PCDAI or wPCDAI in pediatric patients. - Endoscopic remission or mucosal healing: assessed by endoscopic study and using endoscopic scoring indices such as the CDEIS or SES-CD - Mucosal healing: visu or by SES-CD index (SES-CD = 0). - Analytical response of remission: with inflammatory markers such as C-reactive protein, erythrocyte sedimentation rate or fecal calprotectin. - Anthropometric assessment and nutritional status: weight, height, body mass index, insulin-like growth factor type 1, albumin, prealbumin. - Adherence and tolerance to treatment.

CD, Crohn’s disease; PEN, partial enteral nutrition; EEN, exclusive enteral nutrition; CDAI, Crohn’s disease activity index; HBI, Harvey–Bradshaw index; PCDAI, Pediatric Crohn’s Disease Activity Index; wPCDAI, Weighted Pediatric Crohn’s Disease Activity Index; CDEIS, Crohn’s Disease Endoscopic Index of Severity; SES-CD Simple Endoscopic Score for Crohn’s Disease.

**Table 2 nutrients-14-05263-t002:** Inclusion and exclusion criteria of the included studies.

Reference (Year, Country)	Inclusion Criteria	Exclusion Criteria
**A. Levine et al.,** **2019** [31] **Canada,** **Israel**	- PCDAI > 10 and ≤40 and elevated inflammatory markers: CRP > 2 g/L, ESR > 20 mm/h, fecal calprotectine > 200 mcg/g. - <36 months since diagnosis.	- Recent use of steroids. - Recent initiation or dose adjustment for IM. - Past or current biologics’ use. - Primary colonic disease with significant rectal involvement. - Active perianal disease.
**D. Urlep et al.,** **2019** [34] **Slovenia**	- Clinically and endoscopically active CD. - PCDAI > 10. - SES-CD > 3. - Age ≤ 18 years. - No changes in maintenance treatment in the last 3 months.	- PCDAI ≤ 10. - SES-CD ≤ 3. - Changes in maintenance treatment. - Steroids in the last 3 months. - Penetrating disease. - Active perianal disease. - Extra-intestinal disease. - Fixed strictures or small bowel obstruction.
**C. Wall et al.,** **2018** [35] **New Zeland**	- Active CD defined as active disease visible by endoscopy or radiology or an elevated fecal calprotectine. - CD involving at least the ileum.	- Isolated colonic disease. - Active psychological illness. - Corticosteroids in the last fortnight.
**R. Sigall-Boneh et al.,** **2020** [32] **Israel, Canada**	- PCDAI > 10 and ≤40 and elevated inflammatory markers: CRP > 2 g/L, ESR > 20 mm/h, fecal calprotectine > 200 mcg/g. - <36 months since diagnosis.	- Recent use of steroids. - Recent initiation or dose adjustment for IM. - Past or current biologics use. - Primary colonic disease with significant rectal involvement. - Active perianal disease.
**Yanai H. et al.,** **2022** [33] **Israel**	- Established ileal or ileocolonic CD - Disease duration < 5 years. - Active non-complicated mild-to-moderate disease defined by: ▪ HBI 5–14. ▪ Ileocolonoscopic or inflammatory finding on imaging. ▪ Elevated inflammatory markers (positive video capsule test, CT enterography, magnetic resonance, CRP > 5 mg/L or fecal calprotectin > 200 μg/g).	- Structuring or penetrating phenotype (B2/B3). - Active extra-intestinal, perianal disease, deep ulcers involving the distal colon, previous intestinal resection. - Use of IM or dose change in the previous 8 weeks. - Current or past use of biologics. - Systemic steroids/>3 mg budesonide. - Positive stool cultures or C. difficile. - Pregnancy or lactation. - No consumption of protein from animal sources.

CD, Crohn Disease; PCDAI, pediatric CD activity index; CRP, C-reactive protein; ESR, erythro-cyte sedimentation rate; HBI, Harvey–Bradshaw index; IM, immunomodulators; SES-CD, simple endoscopic score for Crohn’s disease.

**Table 3 nutrients-14-05263-t003:** Characteristics of the included studies.

Reference (Year, Country)	Population *n* (Age)	Intervention (Total Weeks)	Formula Employed	Diet	Gender ^1^ and Age ^2^			Location ^1^ (Paris)		
**A. Levine** **et al.,** **2019** [31] **Canada,** **Israel**	78 (4–18 years)	- **Group 1**: EEN (6 weeks) followed by PEN 25% + CDED (6 weeks). - **Group 2**: PEN 50% + CDED (6 weeks) followed by PEN 25% + CDED (6 weeks).	Modulen^®^	CDED		**EEN group** **(*n* = 40)**	**PEN group** **(*n* = 34)**		**EEN group** **(*n* = 40)**	**PEN group** **(*n* = 34)**
					**Female, *n* (%)**	14 (41%)	14 (35%)			
								L1	14 (41%)	18 (41%)
					**Age, (years)**	14.5 ± 2.6	13.8 ±2.8	L2	1 (2.9%)	2 (5%)
								L3	15 (34%)	19 (47%)
								L4a	13 (38%)	14 (35%)
								L4b	3(8.8%)	2 (5%)
**D. Urlep** **et al.,** **2019** [34] **Slovenia**	22 (<25 years)	- **Group 1**: EEN (6 weeks). - **Group 2**: PEN 75% + 1 meal per day of CDED (6 weeks).	Alicam^®^ (Nutricia)	AID-CD		**EEN group** **(*n* = 11)**	**PEN group** **(*n* = 11)**		**EEN group** **(*n* = 11)**	**PEN group** **(*n* = 11)**
					**Female, *n* (%)**	8 (73%)	5 (46%)	L1	0	1 (9%)
								L2	4 (36%)	4 (36%)
					**Age, (years)**	13.8 (3.6–18)	13.4 (9.8–17.9)	L3	7 (64%)	6 (55%)
								L4a	9 (82%)	10 (91%)
								L4b	1 (9%)	3 (27%)
								L4ab	1 (9%)	2 (27%)
**C. Wall** **et al.,** **2018** [35] **New Zealand**	38 (16–40 years)	- **Group 1**: EEN (8 weeks). - **Group 2**: EEN (2 weeks) + PEN with 1 meal per day (6 weeks).	Ensure Plus^®^	Free diet		**EEN group** **(*n* = 25)**	**PEN group** **(*n* = 13)**		**EEN group** **(*n* = 25)**	**PEN group** **(*n* = 13)**
					**Female, *n* (%)**	18 (72%)	12 (92%)	L1	12 (48%)	9 (69%)
								L3	13 (52%)	4 (31%)
					**Age, (years)**	23.3 (15.8 to 38.4)	19.2 (16.5–38.2)	L4	3 (12%)	3 (23%)
								P	2 (8%)	1 (8%)
**R. Sigall-Boneh et al.,** **2020** [32] **Israel, Canada**	78 (4–18 years)	- **Group 1**: PEN 50% + CDED (6 weeks). - **Group 2**: EEN (6 weeks).	Modulén^®^	CDED		**EEN group** **(*n* = 34)**	**PEN group** **(*n* = 40)**		**EEN group** **(*n* = 34)**	**PEN group** **(*n* = 40)**
					**Female, *n* (%)**	14 (41%)	14 (35%)	L1	14 (41%)	18 (41%)
								L2	1 (3%)	2 (5%)
					**Age, (years)**	14.5 ± 2.6	13.8 ±2.8	L3	15 (34%)	19 (47%)
								L4a	13 (38%)	14 (35%)
								L4b	3 (8.8%)	2 (5%)
**Yanai H.** **et al.,** **2022** [33] **Israel**	91 (18–55)	- **Group 1**: CDED alone (24 weeks). - **Group 2**: CDED + PEN (24 weeks).	Modulén^®^	CDED		**CDED group (*n* = 21)**	**PEN group** **(*n* = 19)**	*****	**CDED group** **(*n* = 21)**	**PEN group** **(*n* = 19)**
					**Female, *n* (%)**	13 (62%)	9 (47%)	L1	16 (84%)	19 (90%)
								L3	3 (16%)	2 (10%)
					**Age, (years)**	26 (33 to 38)	34 (25 to 39)	L4	0	2 (10%)

EEN, exclusive enteral nutrition; PEN, partial enteral nutrition; CDED, Crohn’s disease exclusion diet; AID-CD, anti-inflammatory diet for CD; P, perianal. 1. Data are *n* (%), as reported in the corresponding article. 2. Data are median (IR) or mean ± SD, as reported in the corresponding article. * **Montreal classification**.

**Table 4 nutrients-14-05263-t004:** Efficacy of partial enteral nutrition (PEN) in 175 patients with Crohn’s disease in randomized controlled studies.

Reference (Year, Country)	Clinical Response ^1^	Clinical Remission (Week 6) ^1^	Sustained Remission (Week 12) ^1^	Endoscopic Remission ^1^	Analitical Response (CRP Improvement Week 6) ^2^	Calprotectin Decreased (Week 6) ^3^	Compliance/ Tolerance ^1^
**A. Levine et al.,** **2019** [31] **Canada,** **Israel**		**(PCDAI < 10)**		-			
	PEN: 34/40 (85%) EEN: 29/34 (85.3%)	PEN: 30/40 (75%) EEN: 20/34 (58.8%)	PEN: 28/37 (76%) EEN: 14/31 (45%)		PEN: 23.6 mg/dL→5 mg/L EEN: 24 mg/dL→4.1 mg/dL	PEN: −1473 mcg/g (47%) EEN: −948 mcg/g (35.8%)	PEN: 39/40 (98%) EEN: 28/38 (73%)
**D. Urlep** **et al.,** **2019 [34]** **Slovenia**			-	**(SES-CD ≤ 2, at week 6)**			
	PEN: 11/11 (100%) EEN: 10/11 (90.9%)	**(PCDAI < 10)**		PEN 5/11 (45.5%) EEN 5/11 (45.5%)	PEN 16.5→8.8 mg/dL EEN 18.4→7.9 mg/dL	PEN: −288.3 (67%) EEN: −174.2 (45.7%)	PEN 11/12 (91%) EEN 11/13 (85%)
**C. Wall** **et al.,** **2018 [35]** **New Zealand**	-	-	-	-	-	-	PEN 9/11 (81%) EEN: 14/21 (67%)
**Yanai H.** **et al.,** **2021 [33]** **Israel**		**(HBI < 5)**		**SES-CD ≤ 3, at week 24)**			
	PEN 14/19 (74%) CDED 14/19 (67%)	PEN 13/19 (68%) CDED 12/21(57%)	PEN 12/19 (63%) CDED 10/21 (48%)	PEN 8/13 (42%) CDED 6/13 (29%)	PEN 15.8 mg/dL→8.8 mg/dL CDED 12.1 mg/dL→8.2 mg/dL	PEN: +39 mcg/g CDED: −78.5 mcg/g	PEN 16/19 (84%) CDED 13/21 (62%)

PEN, partial enteral nutrition; EEN, exclusive enteral nutrition; CDED, Crohn’s disease exclusion diet; PCDAI, pediatric CD activity index; CRP, C-reactive protein; SES-CD, simple endoscopic score for Crohn’s disease. ^1^ Data are number cases/*n* and rate (%). ^2^ Data are median. ^3^ Data are median and improvement rate (%).

## Data Availability

Not applicable.

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
