# Peer review of "The Role of Partial Enteral Nutrition for Induction of Remission in Crohn’s Disease: A Systematic Review of Controlled Trials"

_nutrients, 2022, doi:10.3390/nu14245263_

Round 1

Reviewer 1 Report

Please see the Attachment.

Author Response

Reviewer 1:

On behalf of all the authors we would like to thank you for your comments and suggestions, they have certainly been very helpful.  We present our reply to each of them separately:

  1. Regarding the first comment, our sincere apologies. Indeed, the first sentence of the discussion is not well referenced. The reference [8] refers to the following sentence (second sentence of the discussion). The bibliographic citation that refers to the first sentence is number [15] "The Medical Management of Paediatric Crohn's Disease: an ECCO-ESPGHAN Guideline Update (van Rheenen, 2020), which has been correctly placed and referenced.
  2. The term analytical result has been replaced in the manuscript as suggested.
  3. Our sincere apologies again for the tables presentation. In order to simplify them we have divided table 2 into two new tables: Table 2: inclusion and exclusion criteria; and Table 3: Characteristics of the included studies. Also, all tables have been polished and restructured to make them easier to read.

We hope that these modifications will meet your demands and facilitate the decision to publish this study.

Reviewer 2 Report

In this systematic review the authors compared the remission of Crohn's disease using Partial Enteral Nutrition and Exclusive Enteral Nutrition. For this, they used 2 databases to search the existing literature with keywords given in the Materials and Methods section. They clearly defined the criteria to select and exclude literature. They included various response rates of patients with CD to PEN and EEN.  

While this review is quite comprehensive it would benefit with inclusion of the following studies that directly deal with the topic in question:

https://pubmed.ncbi.nlm.nih.gov/34339527/

https://pubmed.ncbi.nlm.nih.gov/35679949/

https://pubmed.ncbi.nlm.nih.gov/27401650/

If these studies are not included a justification should be given.

Minor point: Please align the Tables so that they are easier to follow.

Author Response

Reviewer 2:

On behalf of all the authors we would like to thank you for your consideration of this manuscript. We very much appreciate your comments and suggestions, they have certainly been very helpful in improving the manuscript.

The proposed articles are of great interest and quality, and undoubtedly highly relevant to the topic. None of the articles can be included in the systematic review as part of the selected studies since they do not meet the inclusion criteria. Given their great relevance we have decided to include the first two articles in the discussion section of our work. The reason for the non-inclusion of the third article is also explained below.

- Article 1 (https://pubmed.ncbi.nlm.nih.gov/34339527/): Modified Crohn's disease exclusion diet is equally effective as exclusive enteral nutrition: Real-world data. Niseteo T et al. Nutr Clin Pract. 2022 Apr.

This is a real-world retrospective study of the efficacy in induction of remission in children with CD treated with CDED+PEN in comparison with EEN. It shows similar efficacy in both groups with better weight gain in the CDED+PEN group. Our systematic review criteria inclusion are: Interventional analytical studies (randomized or non-randomized clinical trials) published between January 2011 and December 2021, that’s why this retrospective article published in April 2022 cannot be included. Likewise, due to its relevance, we have decided to include it in the discussion section as a new reference [46] (lines 404-407).

- Article 2 (https://pubmed.ncbi.nlm.nih.gov/35679949/) : Metabolome Changes With Diet-Induced Remission in Pediatric Crohn's Disease. Ghiboub M et al. Gastroenterology 2022 Oct.

This randomized clinical trial is an interesting recently published study that characterizes the changes in fecal metabolites induced by CDED+PEN and EEN and their relationship with remission. Unfortunately, this study was published later than our time criterion for inclusion. Although, we consider that this study supports the theory that dietary therapy may play a role in achieving remission as we postulate in our discussion section. It is referenced as [45] in lines 383-390.

Article 3 (https://pubmed.ncbi.nlm.nih.gov/27401650/): Therapeutic Efficacy of Oral Enteral Nutrition in Pediatric Crohn's Disease: A Single Center Non-Comparative Retrospective Study. Kim HJ et al. Yonsei Med. 2016 Sep.

This is a retrospective single-center non comparative study on the efficacy of EEN and PEN in the induction and maintenance of remission in pediatric patients with Crohn's disease. Undoubtedly, this is an interesting study we had already considered during the bibliographic search strategy, and it was excluded by abstract review as it does not meet the inclusion criteria since it is a retrospective study, without a control group. It has not been mentioned in the discussion either since the main intervention of the study is not the subject of our systematic review (comparing EEN vs PEN as induction therapy in CD). This study analyses the effect of EEN as induction therapy and PEN as a maintenance therapy. Even so, if you consider it, we are open to consideration of any further comment on our answers.

We hope that the modifications to this article will meet your demands and facilitate the decision to publish this study.

Reviewer 3 Report

The manuscript scrutinized controlled trials related to the effectiveness of enteral nutrition for induction of remission in Crohn's disease. Authors compared partial enteral nutrition to exclusive enteral nutrition as first line therapy. The research is an interesting, well performed analysis. 

Only tables 2 and 3 are hard to read and should be corrected.  

Author Response

Reviewer 3:

On behalf of all the authors we would like to thank you for your consideration of this manuscript. We very much appreciate your comments, they have certainly been very helpful in improving the manuscript.

Our sincere apologies again for the tables presentation. In order to simplify them we have divided table 2 into two new tables: Table 2: inclusion and exclusion criteria; and Table 3: Characteristics of the included studies. Also, all tables have been polished and restructured to make them easier to read.

We hope that these modifications will meet your demands.
